# The Application of Biomaterials in Spinal Cord Injury

**DOI:** 10.3390/ijms24010816

**Published:** 2023-01-03

**Authors:** Chi Feng, Lan Deng, Yuan-Yuan Yong, Jian-Ming Wu, Da-Lian Qin, Lu Yu, Xiao-Gang Zhou, An-Guo Wu

**Affiliations:** Sichuan Key Medical Laboratory of New Drug Discovery and Drugability Evaluation, Luzhou Key Laboratory of Activity Screening and Druggability Evaluation for Chinese Materia Medica, Key Laboratory of Medical Electrophysiology of Ministry of Education, School of Pharmacy, Southwest Medical University, Luzhou 646000, China

**Keywords:** biomaterials, spinal cord injury, central nervous system, tissue engineering, regenerative medicine

## Abstract

The spinal cord and the brain form the central nervous system (CNS), which is the most important part of the body. However, spinal cord injury (SCI) caused by external forces is one of the most difficult types of neurological injury to treat, resulting in reduced or even absent motor, sensory and autonomic functions. It leads to the reduction or even disappearance of motor, sensory and self-organizing nerve functions. Currently, its incidence is increasing each year worldwide. Therefore, the development of treatments for SCI is urgently needed in the clinic. To date, surgery, drug therapy, stem cell transplantation, regenerative medicine, and rehabilitation therapy have been developed for the treatment of SCI. Among them, regenerative biomaterials that use tissue engineering and bioscaffolds to transport cells or drugs to the injured site are considered the most promising option. In this review, we briefly introduce SCI and its molecular mechanism and summarize the application of biomaterials in the repair and regeneration of tissue in various models of SCI. However, there is still limited evidence about the treatment of SCI with biomaterials in the clinic. Finally, this review will provide inspiration and direction for the future study and application of biomaterials in the treatment of SCI.

## 1. Introduction

The spinal cord is an important part of the human central nervous system and plays an extremely important role in the body’s reflex, conduction, motor and regulatory functions. Among the lesions of the spinal cord, spinal cord injury (SCI) is a central nervous system defect caused by traffic accidents, external violence and falls from height. With the continuous development of the global economy, the incidence of SCI shows an increasing trend. In the United States, more than 1 million people suffer from SCI, with more than 12,000 new cases each year [1]. The annual prevalence of SCI in China is 37 per million, with a mean age range of 34.7–54.4 years and a higher prevalence among males than females [2]. SCI not only brings endless pain and pressure to the patient itself but also imposes a serious burden on the patient’s family and society. SCI is difficult to treat and has gradually become a medical problem of global concern.

In recent years, with the continuous development of cell biology, regenerative medicine, and engineering materials science, tissue engineering technology has gradually become a new way to treat SCI. Cells, scaffolds, and growth factors are three key materials for tissue engineering. Cells and growth factors are usually implanted into the scaffolds and then play a therapeutic role in the injury site, while biomaterials are usually used as scaffolds, which can not only satisfy the attachment, delivery and growth of cells but can also mimic the natural extracellular matrix (ECM), which facilitates the binding of cells to tissues [3]. The basic approach is to construct artificial tissues in vitro with different bioscaffold materials, cells, neurotrophic factors, etc., so that the cells can adapt to the internal environment of the organism and the stimulation of biological factors and then be transplanted into the body to repair the damaged tissue [4]. Regarding the application of tissue engineering in the treatment of SCI, we summarize the properties, advantages, and biological activities of various biomaterials in this review. We hope this review provides insights for the future study of biomaterial scaffolds in the treatment of SCI.

## 2. Spinal Cord Injury

Spinal cord injury (SCI) is a serious complication of spinal fractures, including cervical, thoracic, conus medullaris and cauda equina injuries. It is characterized by high morbidity, disability, and mortality [5]. SCI generally manifests as limb sensory, motor, and autonomic dysfunction below the level of injury, which seriously hinders the patient’s daily life [6]. SCI patients with limb paralysis are prone to pulmonary infection, pressure ulcers, abnormal body temperature, etc., and the higher the injury site is, the more prone to complications, such as cardiovascular, respiratory and genitourinary complications, deep vein thrombosis, and chronic neuropathic pain, which may pose a serious threat to the patient’s life [7,8].

The pathogenesis of SCI is divided into primary injury and secondary injury. Primary injury is direct damage that generally has a physical impact on the spinal cord, mainly neuron and glial cell death, nerve axon and glial membrane destruction, blood vessel rupture with hemorrhage, etc. SCI is the main determinant of the severity of the initial extent of damage and the duration of spinal cord compression [9]. When the spinal cord is mechanically damaged, it can cause spinal cord compression, spinal cord contusion and concussion, axonal injury, etc. Primary SCI is a permanent irreversible dysfunction, and there is no effective clinical treatment yet [10]. Secondary injury usually occurs after the primary injury by a variety of factors, and the extent of damage sometimes exceeds the primary injury. Secondary injury is divided into the acute phase, subacute phase, and chronic phase. Acute clinical manifestations include increased cell permeability, apoptotic signal transduction, ischemia, vascular injury, edema, excitotoxicity, ion imbalance, inflammation, lipid peroxidation, and necrotic cell death [9]. Due to the destruction of the blood-spinal cord barrier, inflammatory cells, including macrophages, microglia, T cells, and neutrophils, infiltrate the injured site and trigger the release of many inflammatory cytokines, such as tumor necrosis factor α (TNF-α), interleukin (IL)-1α, IL-1β, and IL-6 [11], leading to organelle damage and oxidative cell death. Chondroitin sulfate proteoglycans (CSPG), as a component of the neuronal extracellular matrix network, are also up-regulated after SCI, which may be related to inflammatory response [12]. The subacute phase is followed by apoptosis, demyelination of surviving axons, Wallerian degeneration, axonal dieback, matrix remodeling, and the evolution of glial scarring around the injury site. The chronic phase will occur six months after SCI and is characterized by cyst formation, axonal retraction, and maturation of glial scars [13]. In addition, SCI can cause the proliferation of astrocytes and microglia in the nervous system to form glial scars, change the microenvironment of cell survival, increase proinflammatory cytokines, and imbalance the immune response [14] (Figure 1).

At present, the main treatments for SCI include surgery, drug-targeted therapy, stem cell therapy, biomaterials, tissue engineering scaffolds, and regenerative medicine [15] (Figure 2). Surgical treatment can solve the problems of spinal instability and fracture fragment displacement caused by primary injury. That is, surgery reduces the displaced vertebral body or removes the fracture fragments that protrude into the spinal canal to maintain the stability of the spine or to relieve the fracture to minimize further damage [16]. Dimar et al. used adult male rats to conduct decompression surgery in the early stage of SCI. The results showed that surgery can reduce SCI and is conducive to the recovery of neurological function [17]. In addition, Li et al. found that the paraspinal muscle approach combined with the posterior median approach effectively treats thoracolumbar fracture combined with SCI. This treatment has the advantages of a short operation time, minimal trauma, less bleeding and rapid recovery of patients after the operation [18]. Surgical treatment also has some defects. Patients have to bear postoperative physiological pain, and generated wounds due to surgery can easily damage the physiological environment of the body and cause infection. Early implementation of decompressive surgery has limited efficacy in complete SCI. In addition, many drugs, including hormone drugs, gangliosides, nerve growth factors, and traditional Chinese medicine (TCM), are also often used for the treatment of SCI. Among them, methylprednisolone (MP), a synthetic glucocorticoid, inhibits SCI-induced apoptosis by reducing the expression of the low-affinity nerve growth factor receptor p75 (p75NGFR) [19]. Studies have shown that gangliosides have a protective effect on damaged nerve cells and reduce further pathological changes after nerve damage via inhibiting apoptosis [10]. Nerve growth factor (NGF) can promote nerve cell differentiation and survival and axonal growth, which promotes the repair of damaged neurons, maintains the survival of mature neurons and promotes the regeneration of axons [20]. Hormonal drugs are prone to adverse reactions, and excessive doses cause other complications. The ingredients of TCS are complex and the mechanism of action is not clear. After SCI, a large number of axon cells and glial cells are lost and glial scarring prevents cell regeneration. Emerging evidence indicates that stem cell therapy has the characteristics of nerve regeneration, neuroprotection, and immune regulation, so stem cell therapy is also considered to be one of the most promising treatments in regenerative medicine [21]. To date, many stem cell types, including mesenchymal stem cells, olfactory ensheathing cells (OECs), Schwann cells (SCs), oligodendrocyte progenitor cells (OPCs), neural stem cells (NSCs), embryonic stem cells (ESCs), and induced pluripotent stem cells (iPSCs), have been intensively studied and tested in clinical trials [22]. Experimental studies have shown that exosomes derived from adipose-derived mesenchymal stem cells (ADMSCs) change the polarization direction of macrophages in rats with SCI and inhibit the formation of glial scarring [23]. The use of Schwann cell (SC) implantation into the injured spinal cord reduces cyst formation in the injured tissue, alleviates secondary damage to the tissue surrounding the initial injury site, and moderately improves limb movement [24]. Clinical studies have found that stem cells and drugs can be used to improve the SCI microenvironment to treat chronic SCI, and iPSCs introduced into the chronic SCI site can reduce cavitation and support the survival of transplanted cells [25].Although stem cell transplantation has a wide range of clinical applications, there are problems with stem cell therapy. Cell transplantation may induce lesions in other organs, block blood vessels, induce tumors, and cause rejection in the body, and the mechanism of action of stem cell transplantation is unclear. Thus, it is very important to find the best treatment for SCI. Recently, biomaterials have been used as carriers of cells or drugs due to the unique physicochemical properties, providing more treatment options for SCI.

Compared with other treatment methods, biomaterials are of great value in treating and repairing damaged tissues and assisting in the delivery and release of drugs. Drug delivery through biomaterials can reduce the problems of large invasion and blockage of direct drug administration. Although there are many biomaterials used in SCI in preclinical experiments, there are few generalizations about the characteristics of biomaterials and few review studies about the application of biomaterials in SCI. Therefore, this article reviews various types of biomaterials, focusing on various material properties, treatment modalities, etc. It provides a reference for biomaterials in the clinical treatment of SCI and provide directions for further scientific research.

## 3. The Application of Biomaterials in SCI

Biomaterials has been an emerging discipline in recent years, the study of which involves materials science, chemistry, medicine, pharmacology, etc. Biomaterials, also known as biomedical materials, are a class of natural or synthetic materials that can diagnose, repair, treat and induce regeneration of cells, tissues and organs and function in contact with body tissues or body fluids while causing fewer adverse reactions to the organism. Thus, biomaterials are widely used in clinical medicine, biotechnology and many other fields [26]. In the past, biomaterials were mainly used to replace damaged tissues due to diseases or trauma. However, in the last two decades, advances in materials manufacturing and characterization, the evolution of medical regeneration strategies and breakthrough successes in cellular and molecular biology and genetics have provided the substrate for the development of novel biomaterials with innovative applications. Emerging biomaterials aim not only to restore the structure and function of damaged tissues but also to regenerate them through active and targeted interactions [27]. Current biomaterials have been developed over a long period of time, as evidenced by the use of shells and gold for teeth and the use of twine for wound closure in the past, to the recent emergence of bioceramics, alkali metals, and polymeric materials. After several generations of development in the twenty-first century, biomaterials are now widely used in many fields, such as bone plates, artificial joints, vascular grafts, and nerve conduits [28]. The combination of biomaterials with cutting-edge technologies, such as nanotechnology, 3D printing technology, and neural stem cell technology, has brought biomaterials into the era of intelligent nanobiomaterials. On the other hand, the discipline of biomaterials has become the frontier of biomaterial research in drug delivery, tumor targeting therapy, tissue engineering, nanopreparation, bionic design, etc. [29,30,31]. Biomaterials have a wide range of applications and many varieties and are currently divided into natural biomaterials and synthetic biomaterials. Natural biomaterials mainly include chitosan, hyaluronic acid, fibrin, collagen, etc. [32]. The advantages of natural biomaterials are mainly their excellent biocompatibility, low immunogenicity, and nontoxic degradation [33]. In addition, synthetic materials have emerged and are widely used. In general, synthetic biomaterials, mainly polyethylene glycol (PEG), polylactic acid (PLA), polylactic acid-hydroxyacetic acid copolymer (PLGA), polyacrylamide (PAM), polyvinyl alcohol (PVA), and polymethyl methacrylate (PMMA), are attractive for their strong mechanical properties, customizable structure, and low immunogenicity. Their properties are often easier to tune than those of natural biomaterials. For example, the porosity, stiffness and degradation rates of synthetic biomaterials can be altered to match different types of tissues (Figure 3). The benchmark for the use of biomaterials is mainly their safety and performance. Therefore, biomaterials should have biocompatibility, biodegradability, and chemical stability to avoid rejection between materials and organisms.

### 3.1. Natural Materials

Natural materials are usually available from nature and are nontoxic, most of which include nucleic acids, polysaccharides, proteins, lipids and complex macromolecules. Biological activity can be imparted to materials through the use of natural polymers of nonmammalian and mammalian origin, which can be self-assembled or cross-linked to encapsulate natural tissue properties to form noncytotoxic hydrogels and scaffolds [34] (Table 1).

#### 3.1.1. Hyaluronic Acid

Hyaluronic acid (HA), a widely studied and modified natural polymer for scaffolds, is a polysaccharide composed mainly of neurons and astrocytes, and astrocytes can synthesize high molecular weight HA [66,67]. HA is present in the extracellular matrix and is a major component of the extracellular matrix (ECM), playing an important role in the homeostasis of neuronal tissue by influencing cellular behaviors such as cell migration, value addition, and differentiation [68,69,70]. HA can be cross-linked between polymer chains and form highly tunable scaffolds through a variety of simple chemical modifications [71]. In addition, HA reduces glial scar formation by inhibiting the migration, chemotaxis, and proliferation of lymphocytes. It has been shown that HA affects the value-added of astrocytes, thus playing an important role in harmful glial scar formation, especially with high molecular weight HA [72,73]. After chemical modification and further processing and fabrication, 3D HA scaffolds are generated [74]. Covalently linking hydrogels to HA regulates cell growth and function in vitro, indicating that HA hydrogels modulate cell behavior. For example, astrocytes are affected by HA hydrogels; thus, astrocytes are a common target for HA hydrogel-based therapy [71]. In the treatment of SCI, HA hydrogels significantly reduced the level of glial fibrillary acidic protein (GFAP), indicating the level of active astrocytes, leading to the inhibition of harmful glial scar formation [36,75]. In addition, HA hydrogels improve the therapeutic effect of SCI by transporting drugs and stem cells to the injury area without affecting the cells. Emerging evidence indicates that neural stem cells (NSCs) generate new neurons and glia and secrete neuroprotective factors and growth factors to promote cell repair at the site of injury [76]. When using cell therapy, many cells are diluted or washed away, so keeping the cells at the injury site is the premise of treatment. However, HA hydrogel scaffolds can effectively retain cells in the injury site [77]. To improve the survival of transplanted neural stem cells and the integration of host tissues, adult brain-derived neural stem/progenitor cells (NSPCs) were implanted into hydrogel blends of HA and methyl cellulose (HAMC) modified with recombinant rat platelet-derived growth factor-A (rPDGF-A), which were shown to promote oligodendrocyte differentiation, improve graft survival, and reduce cavitation to improve behavioral capacity. In addition, the interaction between HA and NSCs contributes to the normal function of the adult CNS and the repair of damage [78]. Furthermore, HA hydrogels enhance the survival and expected differentiation of NSCs and improve motor function in SCI rats [35]. Therefore, HA is a useful scaffold that increases the stability and survival of implanted cells to promote SCI repair and improve motor ability.

#### 3.1.2. Collagen

Collagen is one of the most abundant and widely distributed proteins in the human body. It is produced by fibroblasts and is the main component of the ECM, providing tensile strength for tissue growth and stimulating wound healing at the site of injury [79]. Because collagen resembles the ECM in the human body, immune rejection is lower when using biomaterials with composite collagen. In addition, the fibrous structure of collagen provides binding sites to support cell adhesion, migration, differentiation, and proliferation [80,81]. Therefore, collagen has gradually become one of the mainstream biomaterials for the treatment of SCI. In early studies, it was found that collagen acted as a carrier of neurotrophic factors to promote the repair of damaged areas [82]. For example, linear rat tail collagen (LRTC) was used as a carrier for recombinant collagen-bound neurotrophin-3 (CBD-NT3), which was found to restore some motor function and promote axonal regeneration after spinal cord transection damage. Currently, to deliver drugs, cells, proteins, and other substances to SCI injury areas, collagen can be designed into various types of scaffolds, such as sponges and hydrogels, and guide catheters [37]. Breen et al. injected a combination of neurotrophic factor (NT-3) and collagen hydrogel into a hemisected SCI model and found that neuronal and axonal growth was improved, while the inflammatory response and glial scar formation were inhibited [38]. Since collagen alone has poor mechanical strength and thermal stability, it is often used in combination with polymers. Wang et al. reported that collagen-binding vascular endothelial growth factor (CBD-VEGF) promoted axonal regeneration and cardiovascular generation in SCI injury sites [39]. Moreover, the N-calmodulin-modified linearly ordered collagen scaffold (LOCS) was shown to improve the mechanical adhesion of endogenous neural/progenitor stem cells (NSPC) to the scaffold, leading to neuronal regeneration and the improvement of motor function in SCI-transected rats [40]. In addition, the transplantation of LOCS and human placenta-derived mesenchymal stem cells (hPMSCs) improved neuronal regeneration and increased operational function in SCI-transected beagle dogs [41]. In addition, an increasing number of studies have shown that gelatin, a product of partial hydrolysis of collagen, also has therapeutic effects on SCI. It is a large molecule hydrocolloid that is usually combined with stem cells to form a scaffold to treat SCI [83]. Methacrylate-based gelatin (GelMA) hydrogels are very similar to natural extracellular matrixes (ECM) in some basic properties. One study showed that combining GelMA hydrogels with induced pluripotent stem cell (iPSC)-derived NSCs (iNSCs) significantly promotes the proliferation and facilitates functional recovery of neural stem cells [43]. In addition, GelMA can be fabricated into 3D materials, and loading BMSCs and NSCs into 3D GelMA hydrogels provides excellent mechanical properties for stem cell proliferation, migration, and differentiation [44]. Furthermore, GelMA and acrylated β-cyclodextrin can be formed into a supramolecular bioink (SM bioink), which then loads with an O-GlcNAc transferase (OGT) inhibitor and NSCs. The results showed that this scaffold promotes neuronal regeneration and axonal growth in an injury model [84]. Moreover, a sodium alginate/gelatin 3D scaffold encapsulated with neural stem cells and oligodendrocytes significantly improves hindlimb movement and nerve regeneration [45]. In summary, gelatin and collagen is a promising strategy for SCI repair mainly by combining it with stem cells and many neurotrophic factors while designing GelMA as a 3D or hydrogel scaffold material in terms of its role in the promotion of neural stem cell proliferation, migration, and differentiation.

#### 3.1.3. Fibrin

Fibrin is a protein multimer that is mainly derived from plasma proteins and has good biocompatibility [45]. Injection of fibrin into the SCI can fill the lesion and provide a vehicle for implantation of stem cells, drugs and cytokines. Fibronectin can serve as a carrier for individual peripheral blood nuclei, and its implantation into a porcine SCI model promotes tissue repair in the immediate area of injury and restoration of conduction function in the posterior column of the spinal cord [46]. The fibrin hydrogel prepared by thrombin could smoothly implant nasopharyngeal carcinoma cells into the SCI site by providing cell attachment sites and survival signals and effectively alleviate the immune response at the SCI site [85]. To mimic the natural environment of spinal cord tissue, Yao et al. prepared a 3D hierarchically arranged fibrin hydrogel (AFG) that promotes targeted host cell invasion, vascular system reconstruction and axonal regeneration in SCI rat lesions [47]. In addition, promoting the differentiation of human endometrial stem cells to oligodendrocyte progenitors by upregulating miR-219 expression levels and using fibrin hydrogel as a scaffold to deliver human endometrial stem cells to the site of SCI significantly promoted the recovery of motor function in SCI rats [48]. The use of adipose mesenchymal stem cells combined with fibrin matrix reduces the cavity area, enhances tissue retention, inhibits the activation of astrocytes, and improves the microenvironment at the SCI site [86]. He et al. used fibrin hydrogels to deliver exosomes overexpressing VGF, the pro-SCI recovery regulator, and they found that the growth of oligodendrocytes in vivo and in vitro is improved [87]. Therefore, fibrin is usually designed as a hydrogel scaffold, which promotes the regeneration of neural tissue and the recovery of motor function by combining different factors.

#### 3.1.4. Decellularized Scaffold

The decellularized matrix can be made into a porous natural biomaterial scaffold, obtained by removing nerve cells from spinal cord tissue and then freeze-drying them. Decellularized tissue retains the natural ECM components of the spinal cord after decellularization, which contains a large number of signaling molecules that regulate cell development, differentiation, and regeneration [88]. Decellularized tissue removes proteins associated with pathological factors such as the inflammatory response and scar formation and acts as an external microenvironment for implantation into the lesion, optimizing the imbalanced microenvironment after SCI, stimulating axonal regeneration, and promoting injury repair [89]. Cerqueira et al. injected cell-free injectable peripheral nerve matrix (iPN) into the lesions of spinal cord contusions, which successfully established immune tolerance and promoted the survival and axonal growth of SC in the transplanted bodies [49]. Cornelison et al. found that decellularized nerve grafts retain extracellular matrix components such as collagen and glycosaminoglycans, which reduce the ratio of M1:M2 macrophages in a rat cervical contusion model and facilitate axonal growth and nerve tissue repair [90]. It was demonstrated in in vitro experiments that in the structure of a rat spinal cord extracellular matrix scaffold, the decellular scaffold retains type IV collagen, fibronectin, laminin and other components in a three-dimensional meshwork structure with good pore space, which can promote the attachment and proliferation of bone marrow mesenchymal stem cells [91]. Liu et al. showed that decellularized spinal cord scaffolds can be used for polylactic acid–hydroxyacetic acid copolymer microspheres loaded with the pro-axonal growth drug bisperoxovanadium(pic) (bpV(pic)), which promote the viability of cultured neural stem cells and axonal growth by inhibiting phosphatase gene expression and activating the mTORC1/AKT pathway in vitro, as well as accelerate axonal regeneration and functional recovery in rats with SCI [50]. In addition, decellularized scaffolds can also be used to deliver adipose-derived stem cells (ADSCs), which effectively promotes histopathological repair and axonal regeneration, reduces reactive glial cell proliferation, and promotes functional recovery in SCI rats [51]. Therefore, decellularized scaffolds are often used to improve the internal microenvironment of SCI and reduce macrophage infiltration and cavity area.

#### 3.1.5. Chitosan

Chitosan, a polysaccharide derived from chitin, is a positively charged natural biopolymer. Due to its unique properties, such as biocompatibility, biodegradability, adhesion, low toxicity, and the ability to form gels, it is widely used and is reported to have analgesic, antitumor, antibacterial, and hemostatic effects [92,93]. Moreover, the physicochemical properties of chitosan are modifiable, especially by altering and controlling its average acetylation degree (DA). Therefore, chitosan is considered a good biomaterial and has been widely investigated and used in various fields, including drug carriers [94], wound healing agents [95], lung surface active additives, and tissue engineering scaffolds [96]. In traumatic SCI, plasma membrane barrier function is degenerated, and neurons are lost or undergo necrosis [97]. At the same time, the passage of intracellular and extracellular ionic substances through this region leads to rapid conduction block within the nerve fibers. In addition, damage to the membrane may cause secondary damage and produce free radicals, which is accompanied by abnormal molecular signaling, inflammation, immune response, apoptosis, vascular changes, secondary cellular dysfunction, and finally, the production of potent endogenous toxins [98,99]. At the injury site of SCI, glial scar formation, neuronal and oligodendrocyte death, and the upregulation of axonal growth inhibitory factors were detected [100]. Therefore, to avoid the progressive destruction of cells, the barrier function of the damaged membrane needs to be re-established and resorted. Emerging evidence indicates that chitosan can be used to seal the damaged nerve cell membrane, thus serving as an effective neuroprotective agent after acute SCI and as an effective treatment for secondary damage [101]. In vitro and in vivo models of SCI have shown that chitosan is effective in regenerating neural tissue, restoring membrane integrity, and reducing the production of free radicals such as reactive oxygen species (ROS). In addition, autografts combined with biomaterials are a promising idea to allow for increased potential for axonal regeneration and functional recovery after SCI [102]. Chedly et al. reported that fragmented physical hydrogel suspensions (FPHS) containing chitosan and water as fragments had a defined degree of acetylation (DA), polymer concentration, and average fragment size. The implantation of FPHS reduced fibroglial scarring, promoted the reconstruction of the vascular system and spinal cord tissue, and modulated astrocyte responses and inflammatory responses at the site of SCI [52]. In addition, human umbilical cord-derived mesenchymal stem cells (hUC-MSCs) and NT3-containing chitosan scaffolds attenuated the inflammatory response and microglial activation, promoted neuronal regeneration, facilitated neurological recovery, and improved motor function in SCI [53]. A chitosan–gelatin composite scaffold containing HA and acetyl heparan sulfate (HS) enhanced the adhesion and long-term expansion of NSCs and progenitor cells and the differentiation of NSCs and oligodendrocytes to neurons and glial cells [103]. Although great progress has been made in the use of chitosan-based scaffolds in the treatment of SCI, there is still a long way to go for their clinical application.

#### 3.1.6. Alginate

Alginate is a hydrophilic long-chain polysaccharide that exists in the cell wall of algae, and its structure is flexible and gelatinous. Alginate exhibits a gel state when exposed to water, and its gel state can produce a series of 3D structures that facilitate drug transport and wound healing. Its unique chemical composition brings it closer to the physical properties of mammalian ECM than other natural organisms [104]. Therefore, alginate is widely used in the development of axon growth-promoting and targeted drug delivery. Although alginate can promote nerve regeneration under certain conditions, its mechanical strength is insufficient due to the fast degradation rate and a certain immune response. Thus, alginate is commonly used in combination with other natural polymers [105]. Hosseini et al. reported that the encapsulation of neural stem cells into alginate 3D scaffolds inhibited inflammation and reduced lesion size [54]. Covalently cross-linked lyophilized alginate enhanced neural regeneration in the spinal cord, and alginate served as a scaffold to promote regenerative axonal growth and the prolongation of astroglial processes [42]. In addition, alginate was designed as a 3D scaffold to help reconstruct ECM soft tissue at the damaged site by filling the injury-induced cavity. The implantation of a poly-L-ornithine-laminin-alginate (PLO-LAM-ALG) hydrogel scaffold promoted cell migration and slight axonal and nerve axon growth in the presence of cationic peptides [55]. Therefore, alginate is an ideal natural material as a 3D scaffold for the encapsulation of cells and factors in the treatment of SCI.

#### 3.1.7. Agarose

In addition to to sodium alginate and chitosan, agarose is also a natural carbohydrate polymer and has excellent biocompatibility, thermally reversible gel behavior, excellent mechanical properties, and high bioactivity. Therefore, agarose has been widely used in tissue engineering and targeted drug delivery [106,107]. In early studies, different mechanical stiffnesses of agarose gels were found to affect the 3D neural protrusion elongation of sensory ganglia [108]. In subsequent studies, agarose hydrogels could guide 3D cell migration and nerve protrusion growth, and the transplantation of neurotrophic factors into multiluminal catheters promoted axonal regeneration after SCI [109,110]. In addition, templated agarose scaffolds supported linear axonal growth through their uniaxial channels and guided axonal regeneration after SCI [111,112]. Meanwhile, templated agarose could be made into linear arrays for the treatment of the post-damage spinal cord, and templated agarose multichannel scaffolds containing BDNF were reported to enhance nerve regeneration [56,113]. Composite electrodes made of agarose and carbon nanotube fibers (A-CNF) were effective in reducing the activity of astrocytes after the use of anti-inflammatory proteins [114]. In recent studies, the implantation of matrix gel into agarose scaffolds improved motility and promoted cell proliferation and axonal regeneration in SCI [57]. In addition, Yang et al. developed a unique agarose, gelatin, polypyrrole (Aga/Gel/PPy, AGP3) hydrogel, and in vitro results showed that AGP3 induced the differentiation of neural stem cells while inhibiting astrocyte formation and activating endogenous nerve regeneration in the spinal cord, resulting in significant restoration of motor function [58]. Therefore, templated agarose scaffolds serving as natural carbohydrate polymers are useful for the implantation of stem cells and the treatment of SCI.

#### 3.1.8. Nanomaterials

The blood-brain barrier and the blood-spinal cord barrier interfere with the effective concentration of drugs entering the brain to some extent, thus requiring larger doses of drugs to achieve a certain therapeutic effect, which may lead to toxicity and side effects due to the increased concentration of drugs [115]. With unique structures, such as small size, large surface area, and large surface area ratio, nanomaterials can be used as drug carriers to deliver therapeutic drugs to targeted locations, reducing the effect in unwanted areas and improving bioavailability while reducing the occurrence of side effects. In addition, nanoparticles also modulate axonal regeneration and restore signal conduction in the injured spinal cord [116]. For the onset of inflammation secondary to SCI, MP effectively alleviated the inflammatory response and limited the secretion of cytokines at the injury site [117]. However, treatment with MP has significant limitations due to its significant side effects, including thromboembolism, blood pressure changes, ionic disturbances, sepsis, pneumonia, wound infections, and acute corticosteroid myopathy due to its low drug selectivity, high systemic doses, and toxicity [118]. The incorporation of MP into nanoparticles could reduce and avoid these side effects, improve bioavailability, increase the efficiency of drug transport, and improve motor function [59,60]. In a further study, circulating monocytes and neutrophils were internalized by polylactide deglycoside (PLGA) nanoparticles according to their physicochemical properties, which downregulate proinflammatory factors, upregulate anti-inflammatory and regenerative genes, inhibit fibrosis and glial scar formation, promote the regeneration of axons, and improve motor function [61]. Due to their unique characteristics, nanoparticles can be used to improve the mechanical properties of hydrogels, increase surface reactivity, and improve the degree of drug release and bioavailability. The addition of nanoparticles to the hydrogel allows for more effective access to the tissue through the capillaries and more efficient delivery of therapeutic agents to the injury site [119,120]. Nerve tissue and cells have the unique property of generating and transmitting electrical stimuli, which affects not only neuronal firing but also cell proliferation, differentiation, and migration [4]. Therefore, electrical stimulation is a nonchemical method that can be applied to nerve regeneration. One of the most important aspects for the use of electrical stimulation to repair spinal cord injuries is the need to find a biocompatible electroactive biomaterial. Numerous studies have shown that the carbon-based nanomaterial graphene has excellent physicochemical and mechanical properties and good electrical conductivity, which allows it to use neural electrical signals in spinal cord tissue to stimulate axonal regeneration and promote the differentiation of stem cells. Graphene is a 2D material consisting of a single layer of carbon atoms arranged in a hexagonal honeycomb lattice, which has excellent properties such as high carrier mobility, quantum Hall effect at room temperature, high optical transparency, excellent mechanical strength, and excellent thermal conductivity [121,122,123]. Thus, graphene has a wide range of applications in various fields, such as sensors [124], supercapacitors [125], composite materials, cancer targeting, photothermal therapy, drug delivery, and tissue engineering. In addition, due to its large surface area and its ability to be surface functionalized, graphene can also be used as a nanocarrier to transport drugs, proteins, etc. Among them, graphene oxide (GO), a member of the graphene family, is widely used for the treatment of SCI because of its large loading, hydrophilic functional groups and excellent biocompatibility. In a recent study, GO was used in combination with polyethylene glycol (PEG) and chitosan (CS) to form novel nanocomposites, which promoted cell growth, reduced inflammatory responses, and improved motor function [62]. Three-dimensional graphene, also known as graphene foam, has been well used in tissue repair. Studies have shown that 3D graphene with better biocompatibility can stimulate and accelerate the growth and differentiation of nerve cells in axonal repair and inhibit neuroinflammation [126,127]. In addition to classical graphene nanoparticles, nanoparticles such as metal nanoparticles and polymer nanoparticles also exist. Metallic nanoparticles are currently showing the potential to design novel delivery systems, which can be divided into pure metal nanoparticles and metal oxide nanoparticles. They act on SCI by changing their shape and size, and are then modified by various types of chemical functional groups. Subsequently, modified metal nanoparticles can bind various drugs, antibodies, nutritional factors, etc. [128,129]. Zonisamide (1,2-benzisoxazole-3-methanesulfonamide) has been reported as an antiepileptic drug, but some studies have found that it can exert a certain therapeutic effect on neurological dysfunction. In Fang’s study, the use of zonisamide-loaded metal nanoparticles showed promise in modifying neurons and axons to promote recovery from SCI [130]. In addition, metal nanoparticles can also promote the immunogenicity of protein immunity. In one study, gold nanoparticles were used as adjuvants to enhance the activity of a 15 nm GNP-coupled human NgR-Fc (hNgR-Fc) protein vaccine, thereby promoting damage repair [131]. Another study showed that laser exposure of gold nanorods can promote the growth of reaching elements [132]. Therefore, in Mina’s study, the authors combined chondroitinase ABCI (cABCI) with different concentrations of gold nanorods. The results of the study exhibited better stability of the enzyme upon binding, thus reducing chondroitin sulfate proteoglycans (CSPG) and promote neuronal regeneration [133]. Moreover, polymeric nanoparticles have also been shown to be therapeutically effective in spinal cord injuries. Ling et al. investigated whether that combining poly(α-lipoic acid)-methylprednisolone (PαLA-MP) prodrug nanoparticles (NPs) and minocycline (MC) could produce better anti-inflammatory effects [134]. In addition, combining rapamycin with mesoporous polydopamine nanoparticles has good ROS clearing ability and exhibits reduced injury cavity, enhanced motility and promotes nerve regeneration in animal models [135]. Taken together, nanomaterials play a key role in carrying and transporting drugs or cells to the targeted tissue of SCI.

#### 3.1.9. Self-Assembled Peptides

Self-assembled peptides (SAPs) are monomers composed of short or repetitive amino acid sequences that, when assembled, can form nanostructures that are attractive in the field of regeneration. By modifying their physicochemical properties and amino acid composition, SAPs perform a variety of biological functions and are more reactive than other conventional nonbiological materials. When composed of nano- or micro-meter structures, SAPs exhibit various abilities, such as tissue regeneration engineering, drug delivery, stability and targeting of drug release, and reduction of toxicity and side effects. The advantages of SAPs include biocompatibility, ease of synthesis, and ability for effective target recognition [136]. However, due to their poor metabolic stability and rapid clearance, SAPs are not always suitable as drugs [137]. In SCI, self-assembling peptide nanofiber scaffolds (SAPNS) can bridge the injured spinal cord, trigger axonal regeneration, inhibit the inflammatory response, and ultimately promote the recovery of motor function [138]. RADA(16)-IKVAV self-assembled peptide hydrogels were used as functional peptide-based scaffolds to load neural stem cells, which increased neural axon growth and reduced glial astrocyte formation [139]. In addition, self-assembling peptide gel (SPG-178) promoted the expression of nerve growth factor (NGF), brain-derived neurotrophic factor (BDNF), neurotrophic factor-4 (NT-4), and promyosin receptor kinases (TrkA and TrkB) in nerve synapses and inhibited neuroinflammation and glial scar formation [140]. Zhao et al. proposed that the combined injection of neural precursor cells and SAPs into damaged neural tissue could enhance neural repair and regeneration [63]. Wang et al. prepared a new FGLmx peptide hydrogel scaffold using pure RADA16 and found that it promoted the proliferation and migration of neural stem cells, thereby facilitating the repair of SCI [64]. In addition, the transplantation of microvessels into RADA-16 l reduced inflammation and glial scar formation and increased the density of axonal growth to the injury/graft site, which demonstrates the potential of vascularized scaffolds in the repair of SCI [65]. Therefore, although many studies have demonstrated the therapeutic role of SAPs, more research is required to increase their metabolic stability and reduce their clearance rate.

### 3.2. Synthetic Materials

Synthetic materials are widely used in tissue engineering and regenerative medicine for the treatment of SCI because they mimic the physicochemical and mechanical properties of cellular tissues. In addition, synthetic materials have low toxicity to the CNS and are reported to inhibit the inflammatory response and reduce glial scarring. Furthermore, the degradation mode of synthetic materials is simple hydrolysis, which facilitates their applications in tissue engineering [141,142]. However, the hydrolysis process has certain drawbacks in that it produces carbon dioxide, which lowers the local pH value and thus leads to cell and tissue necrosis, so more care should be taken when using these materials [143]. Among these synthetic materials, aliphatic polyesters are biodegradable materials, mainly including polylactic acid (PLA), polyglycolic acid (PGA), polycaprolactone (PCL) and their copolymers. They are well known for their high mechanical strength, flexibility, easy processing, and nontoxic degradability [144,145]. In addition, the blending of different types of polymers can form a unique scaffold material with its original properties and exhibit good therapeutic effects in various models, which has been a research direction of biomaterials (Table 2).

#### 3.2.1. Polyethylene Glycol

Polyethylene glycol (PEG) is a nonionic water-soluble polymer that is a hydrolysis product of ethylene oxide and exhibits different physical properties depending on its molecular weight, which is widely used in food and drug applications as a solvent, surfactant, solubilizer, etc. [171]. In addition, PEG can reduce the adverse reaction of drugs by prolonging the duration of action and reducing the immunogenicity of drugs [172]. Thus, PEG is considered to be a safe and nontoxic polymer. When PEG is used in combination with hydrogels, the different molecular weights of PEG affect the mechanical properties of the hydrogels [171,173]. Currently, PEG is usually combined with nanoparticles to form PEGylated nanoparticles, which increase the circulation time, facilitate crossing the blood-brain barrier, and reduce the adverse effects of drugs [174,175]. A composite scaffold consisting of cell growth factor 2 (FGF2), epidermal growth factor (EGF), glial-derived neurotrophic factor (GDNF), PEG, and PCL significantly improved motor function and increased axonal regeneration [146]. In addition, PEG cross-linked with hydrogels has been widely used to improve the regeneration of axons after SCI due to its hydrophilicity, high water content, porosity [176], rejection of nonspecific protein adsorption and cell adhesion [177,178]. In addition, PEG hydrogels in a 3D environment can be used as drug delivery carriers to modulate inflammatory responses, promote axonal regeneration, facilitate molecular and cellular uptake and diffusion, and promote the differentiation of stem cells [179]. Loss of cell membrane integrity leads to cell death and the spread of damage to parts beyond the injury. In SCI, PEG inhibits free radical generation and resists the level of lipid peroxidation, indicating that PEG acts as a fusion agent to reseal the disrupted plasma membrane and protect mitochondria against injury [147]. Therefore, the key application of PEG serves as a fusion agent to repair the cell membrane at the injury site, leading to a reduction in oxidative stress, axonal regeneration, and the restoration of motor function [148,180]. However, it has been found that PEG is only effective in repairing cell membranes at higher concentrations, while 1,2-distearoyl-sn-glycero-3-phosphoethanolamine-N-[methoxy(polyethylene glycol)-2000] (DSPE-PEG) could repair the dysfunction of SCI more effectively at lower concentrations [149]. Therefore, PEG is an ideal biomaterial that inhibits the disruption of the cell membrane to promote the regeneration of neurons in SCI.

#### 3.2.2. Polylactic Acid

Polylactic acid (PLA) is made from renewable resources and is the most commonly used and promising renewable aliphatic polyester for the synthesis of biodegradable polymers [181]. PLA is now widely used in biomedical applications, such as tissue engineering scaffolds, drug delivery systems, implants, and coverings. In addition, unique properties, such as nontoxicity, bioresorbability, environmental friendliness, and thermoplasticity, make PLA and its lysis products biocompatible with SCs and spinal cord tissue [182,183]. Therefore, PLA can be designed into various types of scaffolds for the regeneration of tissue cells after SCI, including micron or nanofibers, hydrogels, microporous sponges, etc. [184,185]. However, PLA also has certain disadvantages, including poor toughness, slow degradation, and hydrophobicity [186,187], so it is usually used in combination with other polymers or nonpolymers. For example, PLA/PPy scaffolds formed by polylactic acid (PLA) and polypyrrole (PPy) composite could aid the transplantation of bone marrow stromal cells (BMSCs) in SCI, leading to the regeneration of neurons, the restoration of conduction, the inhibition of glial scar formation, and the regeneration of axons, which provides a better microenvironment for the treatment of SCI [150]. In addition, drug-loaded fiber mats made of core-shell nanofibers equipped with docosahexaenoic acid (DHA) in the core and PLA as the shell were prepared by electrostatic spinning, which has sufficient mechanical properties to promote the growth of neuronal synapses [151]. In another study, aligned PLA microfiber grafts without any cells, neurotrophic factors, or drugs promoted the regeneration of CNS tissues [152]. There, PLA has a wide range of application aspects and can be designed into different scaffold types to treat SCI.

#### 3.2.3. Poly(Lactic-Co-Glycolic Acid)

Poly(lactic-co-glycolic acid) (PLGA) is formed by the polymerization of lactic acid and ethanol, which are degradable functional organic polymers. PLGA has different molecular weights and copolymer ratios, indicating that the ratio of the two monomers affects their physicochemical properties [188]. Studies have shown that PLGA has the following properties: slow controlled release, low toxicity, biodegradability, and biocompatibility [189]. Thus, PLGA can be used as a carrier for hydrophobic and hydrophilic substances [190,191]. To date, PLGA hydrogels, catheter scaffolds, and nanoparticles are the three most widely used drug delivery systems [192]. PLGA is usually used in combination with nanoparticles, and PLGA nanoparticles can increase the retention time of macromolecules in vivo and regulate the release rate and distribution of macromolecules [193]. Gwak et al. reported that PLGA nanospheres modified with 3β-[N-(N′,N′-dimethylaminoethane) carbamoyl] cholesterol PLGA/DC-Chol nanospheres improved the efficiency of gene transfection, promoted vascular and tissue regeneration, and improved the motility of SCI rats [153]. Moreover, PLGA/PLLA has been used as a composite scaffold to treat SCI in several ways. For example, the implantation of human oral mucosal stem cells (hOMSCs) into the PLGA/PLLA scaffold could promote endogenous repair by secreting neuroprotective, immunomodulatory, and axonal extension-related factors, thereby restoring motility and reducing glial scar formation [154]. In addition, PLGA/PLLA scaffolds implanted with human dental pulp stem cells (DPSCs) are highly vascularized and have great potential for tract vascular production and nerve regeneration in SCI rats [155]. It has been reported that the diameter and porosity of scaffolds have a significant impact on axonal regeneration of the damaged spinal cord, with smaller diameter scaffolds showing a better effect for SCI repair [194,195]. Therefore, designing PLAG multichannel catheters with hierarchical pore structures can enhance gene expression in vivo and promote value-added cell spreading, which can improve the repair of damaged spinal cords [196]. In addition, longitudinal porous scaffolds prepared by mixing HA hydrogels and PLGA have a better therapeutic effect on SCI. For example, the combinational use of anti-Nogo receptor antibodies (antiNgR) and PLGA microspheres containing BDNF and VEGF could inhibit inflammation and promote revascularization [156]. Furthermore, PLGA scaffolds can promote the differentiation of stem cells in the tissue of SCI [197], and the implantation of NSCs into PLGA scaffolds can effectively promote the recovery of neurological functions in vivo [198,199]. In conclusion, the development of PLGA scaffolds and their composite scaffolds with other polymers has become popular and widely used in the treatment of SCI, and more research on gene modification and optimization of the physical properties of the scaffold are required in the future.

#### 3.2.4. Polycaprolactone

Polycaprolactone (PCL) is a highly elastic fatty acid polyester that is biocompatible and biodegradable, and it is commonly used in tissue engineering because of its slow degradation rate and low toxicity [200,201]. In a recent study, 3D-aligned PCL microfibrous scaffolds of different diameters were prepared, and their effects on the differentiation of stem/progenitor cells were investigated. The results showed that the differentiation of stem cells, astrocytes, and oligodendrocytes was increased by PCL scaffolds, suggesting that PCL is useful for the treatment of SCI [202]. The implantation of differentiated cells from hEnSCs cocultured with human Schwann cells (hSCs) onto PCL/gelatin scaffolds limited secondary damage, promoted axonal regeneration, and restored sensory and motor function after SCI [157]. To allow PCL to function as a nerve-guided scaffold with maximum open volume, the cell adhesion of PCL is improved by changing the porosity of PCL, resulting in the fabrication of porous PCL microtubular scaffolds, which have been demonstrated to promote axonal growth and reduce scar tissue in SCI rats [159]. In addition, Gelain et al. found that scaffolds containing PCL nanofibers and bioactive self-assembling peptides induced neural regeneration and angiogenesis and restored the motility of SCI rats [203]. Furthermore, a trimethylene carbonate and ε-caprolactone (TC) block copolymer catheter scaffold containing longitudinally microgrooved poly-p-dioxanone microfilaments (PDOs) was designed, and the results showed that the TC/PDO catheter promoted the migration of different cell types, such as SCs, while promoting the growth of sensory axons and attenuating the response of astrocytes to SCI injury [158]. Therefore, PCL scaffolds promote cell differentiation and improve the motor function of SCI.

#### 3.2.5. Polyvinyl Alcohol (PVA)

Polyvinyl alcohol (PVA) is a nontoxic, biocompatible, and biodegradable hydrophilic polymer. In the treatment of SCI, PVA was found to significantly inhibit inflammatory responses and alleviate secondary damage. In an early study, the implantation of PVA hydrogel into the spinal cord prevented the migration of inflammatory cells and reduced the formation of scar tissue and adhesions [204]. In a subsequent study, PVA and PVP containing p38 and JNK inhibitors reduced the inflammatory response and decreased the expression of proteins associated with cell death [160]. In addition, Chen et al. prepared a composite molybdenum sulfide/graphene oxide/PVA (MoS 2/GO/PVA) hydrogel with good biocompatibility, electrical conductivity, and moderate mechanical properties. They found that this hydrogel could promote the differentiation of neurons, inhibit the inflammatory response, and gradually restore motor function [161]. In conclusion, research on PVA has mainly focused on its anti-inflammatory effect in SCI, and more research on the biological activity of PVA is needed.

#### 3.2.6. Poly-Beta-Hydroxybutyrate

Poly-beta-hydroxybutyrate (PHB) is a biopolyester with a large molecular weight that exists in the cytoplasm of many bacteria and is a degradable biomaterial [205]. The transplantation of SCs into PHB catheters could better fill the cavity and significantly increase SC cell attachment, proliferation, and axon regeneration [206]. In recent years, Zhao et al. found that the blending of polyhydroxybutyrate-cohydroxyvalerate (PHBV) with PLA and collagen (Col) could improve the performance of the polymer. The prepared nanofiber scaffold significantly promotes the differentiation of astrocytes but inhibits their overactivation. It can also reduce CSPG and glial fibrillary acidic protein (GFAP) and promotes the recovery of motor function [162]. Another study showed that the implantation of PHBV/PLA/Col membrane and VSC4.1 motor neurons into rats with SCI can inhibit inflammasomes and reduce glial scar formation by reducing the infiltration of CD86-positive macrophages in the lesion and promoting axon growth [163]. In addition, NLRP3 inflammasome expression can be suppressed using the ketone metabolite β-hydroxybutyrate (βOHB), resulting in less neuroinflammation and better motor performance and electrophysiological recovery [164]. Doncel-Pérez et al. used poly(3-hydroxybutyrate-co-3-hydroxycaproic acid) fibrous scaffolds coated with laminin or polylysine/laminin to culture rat neural progenitor cells. Then, synchrotron radiation infrared microspectroscopy observation results showed that the neural progenitor cells had good adhesion and proliferation behavior [207]. Agrawal et al. developed a new type of biomaterial mixed melanin with PHB. This scaffold is compatible with nerve tissue in terms of physicochemical and electrical properties, and has rich surface nanotopography, semiconducting properties, and pore viscoelastic properties similar to brain tissue. Therefore, it can improve cell adhesion and growth of sensory and motor neurons in mice [208]. Therefore, the research on PHB can start from its mechanism of action and suitable composite materials.

#### 3.2.7. Polysialic Acid

Polysialic acid (PSA), a class of linear homogeneous α-2,8 carbohydrates linked to sialic acid, is most prominent during nervous system development and can be combined with neural adhesion molecules in the vertebrate nervous system through typical n-linked glycosidic bonds [209,210]. PSA can induce the migration of SCs, which has a great effect on improving repair after SCI. Pearse et al. used the lentiviral vector-transduced mouse polysialyltransferase gene ST8SIA4 (LV-PST) and recombinant bacterial enzyme engineering (PSTNm) to upregulate the expression of PSA in nerve cells, proving that PSA can effectively regulate the immune response and improve the migration of SCs and neural repair after SCI [211]. A PSA-based minocycline nanodrug delivery system (PSM) exhibits significant anti-inflammatory and neuroprotective effects in vitro and in vivo, and PSM can recruit endogenous neural stem cells to the injury site in SCI rats, promoting neuronal regeneration and long axon extension throughout the glial scar to improve motor function [212]. Studies have shown that PCL/PSA hybrid nanofiber scaffolds encapsulated with MP implanted into the lesion area can effectively inhibit acute tissue inflammation and apoptosis and promote axon regeneration, thereby promoting nerve repair and improving functional prognosis [165]. In addition, the use of PSA ring mimetic peptide (PR-21) inhibits the hyperplasia of reactive glia and improves motor sensory function [213]. In Mehanna’s study, the combination of PSA and human natural killer cell-1 (HNK-1) glycan shows the poor effect in the early stage of injury but can promote axonal myelination and the recovery of function after 6 weeks [166]. Recently, a collagen laminin (C/L) scaffold was used to load the PSA analog compound 5-nonoxytryptamine (5-NOT), which promotes the survival of cortical neurons and neurite length and improves motor function through the ERK-MAPK pathway [167]. In conclusion, finding a suitable PSA mimic is an important direction for future research on SCI.

#### 3.2.8. Poly(2-Hydroxyethyl Methacrylate)

Most of the current tissue-engineered scaffolds for the treatment of SCI are chosen from biodegradable natural or synthetic polymers because nondegradable or slowly degradable polymers require a postprocessing process, which makes treatments more difficult. Currently, nondegradable polymers such as acrylic acid have been gradually used in the treatment of SCI, of which poly(2-hydroxyethyl methacrylate) (pHEMA) is a flexible hydrophilic porous material with biocompatible and stable mechanical properties that can incorporate collagen and fibrin as a bioadhesive matrix [214]. Tsai et al. used a new synthetic hydrogel composite of PHEMA and methyl methacrylate (MMA) as a guiding channel and found that the PHEMA-MMA hydrogel guiding channel promoted axon regeneration and reduced glial scar formation in spinal cord transection injury [168,170]. In addition, PHEMA hydrogel scaffolds containing BDNF increased the number of axons and blood vessels and promoted recovery of motor function in SCI, accompanied by a mild inflammatory response [215]. A PHEMA hydrogel implanted carrying charged functional groups was reported to improve axonal regeneration at injury sites [216]. PHEMA lacks functional groups that support cell adhesion. After the immobilization of laminin-derived Ac-CGGGASIKVAVS-OH (SIKVAV) peptide and fibronectin subunit (Fn) on PHEMA hydrogel scaffolds, cell adhesion and cell proliferation and differentiation were improved [217]. Kubinova et al. optimized the porosity and mechanical properties of PHEMA hydrogels and found that a 68% porosity of PHEMA hydrogels had the best effect on tissue response and axonal growth in SCI rats [169]. In addition, the design of homogeneous and bilayer PHEMA hydrogel scaffolds has mechanical properties and biocompatibility, which are more suitable for spinal cord tissues [218]. Valdes-Sanchez et al. reported that caprolactone 2-(methacryloyloxy) ethyl ester (CLMA) made into a 3D scaffold after esterification from PHEMA and PCL could improve the survival of mature neurons after binding to acutely transplanted epithelial stem/progenitor cells (epSPCs) [219]. The use of SIKVAV-HEMA hydrogel scaffolds with oriented voids showed better infiltration in connective tissue and blood vessels but slower infiltration in axons, so further research is needed to support the type of scaffold in the microenvironment of the injury site [220]. Therefore, the modification or combinational use of pHEMA with other materials provides therapeutic options for SCI.

### 3.3. Titanium Alloys

Today, titanium and titanium-based alloys are also widely used biomaterials because of their good biocompatibility, low density, corrosion resistance, and good mechanical efficiency. At present, it is mainly used to replace orthopedic implant materials [221]. The supplement of Zr, V, and Mo elements to titanium can improve the mechanical properties of titanium, which manufactures bone plates and cardiovascular stents [222]. Research has shown that mesenchymal stem cells differentiate into osteoblasts and promote bone union and remodeling when four new alloys are implanted into rabbit models. Titanium-based biomaterials with tailored porosity can affect cell adhesion, differentiation, and growth [223]. Therefore, alloy materials can be considered for the treatment of SCI. When titanium-based alloy is used as a biomaterial for SCI repair, it needs to consider that the alloy may react with the liquid environment in the body to produce toxicity and trigger inflammation [224]. Therefore, how to make composite scaffolds from natural or synthetic materials and titanium-based alloys is a promising research direction in the future. 

### 3.4. 3D Scaffold

Three-dimensional bioprinting technology allows the size and shape of the scaffold to be adjusted as needed. Many biomaterials are used to design materials as 3D scaffolds for repairing SCI. In a study, a novel biocompatible bioink consisting of functional chitosan, hyaluronic acid derivatives and matrix gels was found to rapidly gel and spontaneously covalently cross-link to maintain the viability of neural progenitor cells and effectively support axonal regeneration in a rat model of SCI [225]. Three-dimensional-printed scaffolds can transport homogeneous bone marrow MSCs and SCs in a specific spatial arrangement, which promote the formation of intercellular junctions and directed cell differentiation [226]. Cryoextrusion 3D printing technology can maintain the biological activity of the delivered cytokines. The collagen/chitosan scaffold prepared by low-temperature extrusion 3D printing technology successfully fuses brain-derived neurotrophic factors, which fills the injury gap, promotes nerve fiber regeneration, accelerates the establishment of synaptic connections, and promotes myelin regeneration at the injury site in rats with SCI [227]. Three-dimensional-printed collagen/silk fibroin scaffolds which mimic the structure of corticospinal tracts can guide the targeted repair of damaged neural tissue in rats [228]. Joung et al. prepared physiologically active neuronal networks by precisely placing clusters of spinal cord neuronal progenitor cells and oligodendrocyte progenitor cells within 3D printed biocompatible scaffolds during assembly and controlling the cell cluster positions using a distribution-by-dot printing method [229]. Koffler et al. used microscale continuous projection printing to create complex neural network structures and printed 3D bionic hydrogel scaffolds loaded with neural progenitor cells that supported axonal regeneration in SCI lesions in mice, induced neuronal synapses upstream and downstream of the implant to connect to neural progenitor cells within the scaffold, and significantly improved functional outcomes. Moreover, the scaffold is scalable to human spinal cord size and lesion geometry [230]. In summary, the 3D scaffold can fill the damaged part well by adjusting its shape and size.

## 4. Clinical Application of Biomaterial Scaffolds

Although many biomaterial scaffolds have been widely used in the study of SCI, few have been translated from preclinical experiments to clinical studies. Previously, Amr et al. investigated the repair effect of chitosan–laminin scaffold-wrapped MSCs on patients with chronic paraplegia. The results showed that motor function is recovered despite a gap in the spinal cord, the patients regain the ability of some muscle groups, and the thoracolumbar spine is improved. Moreover, the recovery process is not affected by the postoperative seroma formed by the disintegration of chitosan [231]. Xiao et al. studied the safety and feasibility of the collagen scaffold NeuroRegen after implantation into the human body. During the follow-up one year later, they found that there was no adverse reaction related to stent implantation and no more serious consequences such as deterioration of the nervous system. In addition, they observed partial recovery of the patient’s autonomic nervous function [232]. In a recent study, after the implantation of MSCs into NeuroRegen scaffolds, the subsequent follow-up found that the sensory function of the patient’s bowel and bladder recovered and the sensory level expanded below the injury site. In addition, some patients could control the activities of the toes, suggesting the recovery of some muscle tissue [233]. Chen et al. implanted NeuroRegen stents with autologous bone marrow mononuclear cells (BMMCs) into the body. In addition to stress pulmonary ulcer, pulmonary infection and transient high fever in the short term after the operation, no significant cystic cavitation or malignant proliferation was observed in the follow-up 3 years later. In addition, the patients exhibited expansion of the sensory level and recovery of autonomic nervous function [234]. In addition to the collagen scaffold NeuroRegen, a spinal cord scaffold composed of covalently conjugated poly(lactic acid-glycolic acid) and poly(l-lysine) InVivo Therapeutics Corp and CS collagen scaffolds have also been clinically studied [235,236]. These clinical studies have confirmed the clinical feasibility of biomaterials, but most of the current clinical trials remain in phase I studies, and the effectiveness of biomaterials in the treatment of SCI still faces great challenges. It is believed that more scaffold materials will be converted into clinical applications in the future.

## 5. Summary and Future Outlook

At present, SCI has become one of the most difficult central neurological disorders to treat, and patients usually show a decrease or loss of motor and sensory function below the injury site. However, the repair of this damage may be affected by many factors, such as apoptosis of neuronal cells, deposition of glial scarring, and inflammatory cascade response. This article mainly discusses various biomaterial scaffolds for SCI treatment, among which the relatively popular scaffolds are hydrogel and 3D scaffolds. Various materials are designed so that these scaffolds can simulate the internal environment of spinal cord tissue and can meet the repair of the injured part by adjusting the size and shape of the scaffold. Compared with traditional drug intervention and surgical treatment, the use of biomaterial scaffolds can reduce some complex side effects of drugs and obstacles to functional recovery after surgery. The physical and chemical properties of these biomaterials and the structure of the scaffold play an important role in the treatment of SCI. Compared with other treatments, we should pay more attention to the issues of biomaterials, such as biocompatibility, degradability, mechanical strength, and toxicity to peripheral tissues.

Although natural biomaterials mimic ECM and show good biocompatibility, due to their inherent characteristics, there are also some unavoidable shortcomings, such as insufficient scaffold strength, mismatch between cell degradation rate and regeneration rate, and scaffold collapse after the material swells with water. Among them, fibrin has poor mechanical strength and is easy to degrade, which is not conducive to long-term culture in vitro, and the arrangement of fibers lacks directionality, which reduces the induction effect on directional growth of axons. HA is easily soluble in water and absorbs and degrades too quickly. Alginate is prone to the immune response. Synthetic materials increase the hardness of the scaffold but lack inherent biological functions and must undergo significant postprocessing to trigger the desired response in vivo, and their degradation products are prone to local inflammatory responses that destroy the microenvironment, resulting in the decrease in cell survival. In addition, synthetic materials have weak affinity to cells, and are commonly used to prepare composite scaffolds with natural materials to induce nerve axon regeneration. For example, PCL has insufficient mechanical strength; PLA has a slow degradation rate and poor hydrophobicity; PSA has the disadvantage of being difficult to purify; PHB has poor toughness and a narrow processing time window. It is worth noting that regardless of what kind of biomaterial is implanted into the spinal cord, it cannot cause secondary damage to the body.

In conclusion, although these scaffold materials have made some progress in the treatment of SCI, there are still many problems to be solved. When using different scaffolds to treat SCI, the detailed mechanism of action still needs to be further studied, and how to combine the drugs, NTs, and stem cells loaded on the stents to achieve the best therapeutic effect still needs to be deeply understood. In addition, when using a scaffold for treatment, there is uncertainty about the optimal timing of implantation, location, and duration of treatment while the stent is in the body. In future research, the main research directions include the construction of a stable microenvironment, the study of therapeutic mechanisms, how to improve the shortcomings of various biomaterials, and the development and application of new composite biomaterials in order to find materials with the best therapeutic effect. Most importantly, future research should pay more attention to the safety and effectiveness evaluation of biomaterial scaffolds in clinical trials, so as to promote the application of SCI in clinical treatment.

## Figures and Tables

**Figure 1 ijms-24-00816-f001:**
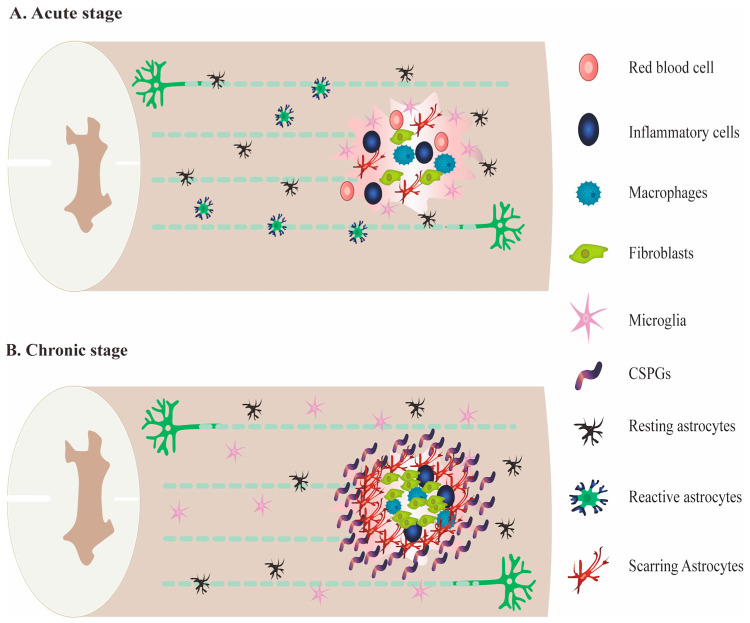
The pathological mechanism in the injury site after SCI. (**A**). Inflammatory cell infiltration and the release of neurotoxins were generated from the inflammatory cascade response of immune cells at the acute stage of SCI, which impeded axonal regeneration in the injury site of SCI. (**B**). Cystic cavities, myelin debris, deposition of GSPGs and astrocytes were progressively generated at the chronic stage of SCI.

**Figure 2 ijms-24-00816-f002:**
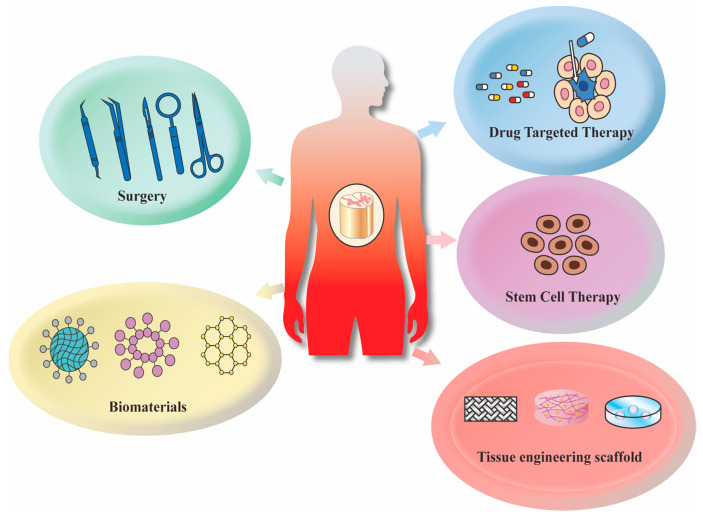
Current treatments for SCI include surgery, drug-targeted therapy, biomaterials, stem cell therapy, and tissue engineering scaffolds.

**Figure 3 ijms-24-00816-f003:**
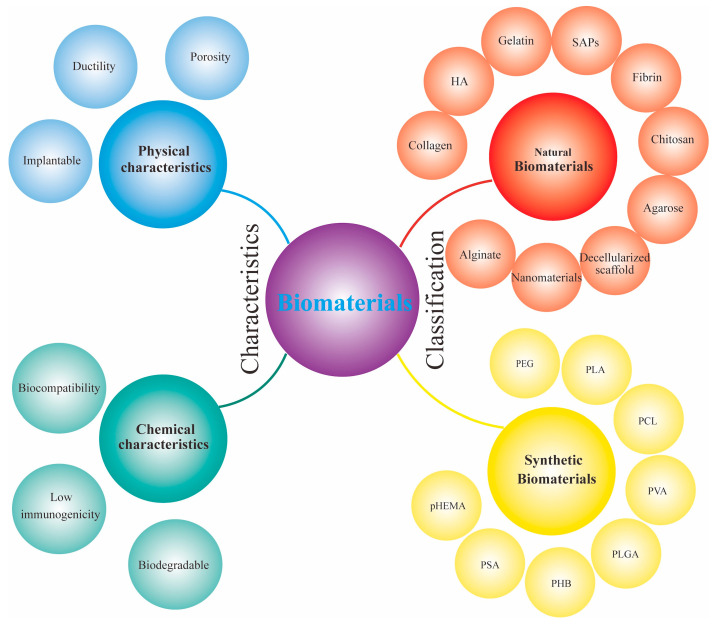
Characteristics and classification of biomaterials.

**Table 1 ijms-24-00816-t001:** The bioactivity of natural biomaterials in various models of SCI.

Natural Biomaterials	Implanted Substance	Animals and SCI Model	Effect	Ref.
Hyaluronic Acid	Human embryonic stem cell-derived neural stem cells	Wister rats with spinal cord injury	Increasing cell differentiation and improving motor function	[35]
Hyaluronic Acid	Nogo-66 receptor antibody and poly-L-lysine	Wister rats with spinal cord hemisection	Promoting vascular regeneration, and inhibiting the formation of glial scar	[36]
Collagen	Linear ordered collagen scaffolds loaded with collagen-binding neurotrophin-3	Wister rats with spinal cord transection	Promoting axonal regeneration and restoring some motor function	[37]
Collagen	Neurotrophin-3	Dawley rats with spinal cord hemisection	Inhibiting inflammation and scar; promoting neuronal regeneration	[38]
Collagen	Implanted vascular endothelial growth factor	Sprague–Dawley (SD) rats with spinal cord transection	Improving the microenvironment and promoting angiogenesis	[39]
Collagen	N-cadherin	SD rats with spinal cord transection	Promoting the regeneration of neurons and improving motor function	[40]
Collagen	Human placenta-derived mesenchymal stem cells	Beagle dog with spinal cord transection	Improving the regeneration of neurons and exercise capacity	[41]
Collagen	Freeze-dried alginate sponge cross-linked with covalent bonds	Rats with spinal cord transection	Enhancing nerve regeneration of spinal cord	[42]
Gelatin	iNSCs	C57BL/6N mice with spinal cord transection	Promoting the proliferation of neural stem cells and significantly promoting functional recovery	[43]
Gelatin	BMSC and NSC	SD rats with spinal cord hemisection	Significantly promoting motor function recovery and neuronal differentiation	[44]
Gelatin	NSC	SD rats with spinal cord transection	Significantly improving hindlimb movement and nerve regeneration	[45]
Fibrin	–	Pot-bellied pigs with spinal cord dorsal column removed	Promoting tissue repair in areas near the injury and restoration of conduction function in the posterior column of the spinal cord	[46]
Fibrin	–	SD rats with spinal cord hemisection	Promoting directed host cell invasion, vasculature remodeling and axonal regeneration	[47]
Fibrin	Human endometrial stem cells (hEnSCs)	Wister rats with spinal cord aneurysm clip	Significantly promoting the recovery of motor function in injured rats	[48]
Decellularized scaffold	iPN and SC	Fischer rats with Moderate thoracic contusion injury	Promoting Schwann cell survival and neurite outgrowth in grafts	[49]
Decellularized scaffold	bpV (pic)	SD rats with spinal cord hemisection	Promoting NSC activity and axon outgrowth	[50]
Decellularized scaffold	ADSC	SD rats with spinal cord hemisecion	Enhancing the ability of axonal regeneration and promoting functional recovery	[51]
Chitosan	–	Wister rats with spinal cord hemisection	Reducing glial scar; improving the inflammatory response	[52]
Chitosan	Neurotrophin-3 and human umbilical cord mesenchymal stem cells	C57BL/6 Mice with spinal cord transection	Promoting the recovery of neurological function and reducing inflammation	[53]
Alginate	Neural stem cells	SD rats with spinal cord Aneurysmal clip	Reducing inflammation and lesion size	[54]
Alginate	Peptides and astroglia	GFP-transgenic F344 rats with spinal cord hemisection	Promoting cell migration and slight axon growth	[55]
Agarose	BDNF	F344 rats with spinal cord transection	Promoting nerve regeneration and axon growth	[56]
Agarose	Matrigel	Sprague–Dawley rats with spinal cord dorsal column removed	Improving motor function and promoting cell proliferation	[57]
Agarose	Gelatin and polypyrrole	Rats with spinal cord hemisection	Inhibiting the formation of astrocytes and activating endogenous nerve regeneration of spinal cord	[58]
Nanomaterial	MP	SD rats with spinal cord hemisection	Reducing lesion volume and improving drug transport efficiency	[59]
Nanomaterial	MP	Wister rats with spinal cord hemisection	Reducing the secondary reaction after SCI	[60]
Nanomaterial	Poly(lactide-coglycolide)	C57/BL6 mice with spinal cord hemisection	Enhancing the expression of anti-inflammatory and regeneration genes and increasing axon regeneration	[61]
Nanomaterial	Chitosan and polyethylene glycol	BALB/c mice with compression injury of spinal cord	Promoting cell growth and reducing inflammatory response	[62]
Self-loading peptide	Neural precursor cells	Wister rats with spinal cord aneurysm clip	Enhancing nerve repair and regeneration	[63]
Self-loading peptide	–	Sprague–Dawley rats with thoracic spinal cord tissues	Promoting the proliferation and migration of neural stem cells	[64]
Self-loading peptide	–	SD rats with injury spinal cord	Reducing inflammation and glial scar formation, and increaseing axonal growth	[65]

–: There is no implanted substance

**Table 2 ijms-24-00816-t002:** The bioactivity of synthetic biomaterials in various models of SCI.

Synthetic Biomaterials	Implantted Substance	Animals and SCI Model	Effect	Ref.
PEG	FGF2, EGF, GDNF	SD Rats with spinal cord transection	Improving motor function and increasing axonal regeneration	[146]
PEG	–	Guinea pig with compression injury of spinal cord	Inhibiting the formation of free radicals, and resistancing lipid peroxidation	[147]
PEG	–	Guinea pig with spinal cord transection	Repairing cell membrane, and reducing oxidative stress	[148]
PEG	DSPE	SD rats with compression injury of spinal cord	Reducing material concentration and improving dysfunction after injury	[149]
PLA	Bone marrow stromal cells	SD rats with spinal cord transection	Promoting nerve regeneration as well as restoration of conduction, and providing a better microenvironment	[150]
PLA	DHA	SD rats with spinal cord hemisection	Promoting axon regeneration with strong mechanical properties	[151]
PLA	Aligned microfiber-based grafts	SD rats with spinal cord transection	Reducing cyst volume in SCI	[152]
PLGA	DC-Chol	SD rats with spinal cord injury	Promoting the regeneration of blood vessels and tissues and improving exercise capacity	[153]
PLGA	HOMSC	SD rats with spinal cord transection	Promoting endogenous repair, thereby restoring exercise capacity	[154]
PLGA	DPSCs	Rats with spinal cord transection	Enhancing the regeneration of blood vessels and axons	[155]
PLGA	AntiNgR	SD rats with spinal cord hemisection	Inhibiting inflammation and promoting angiogenesis	[156]
PCL	HEnSCs and hSC	SD rats with spinal cord hemisection	Limiting secondary reactions and restoring motor function	[157]
PCL	PDO	SD rats with spinal cord hemisection	Promoting axonal regeneration and inhibiting the activity of astrocytes	[158]
PCL	–	Fischer 344 rats with spinal cord transection	Promoting axonal growth as well as reducing scar tissue	[159]
PVA	–	SD rats with spinal cord cervical contusion	Reducing inflammation and reducing the number of cell death—promoting proteins	[160]
PVA	MoS 2/GO	C57BL/6 N mice with spinal cord hemisection	Inhibiting inflammatory and the activation of glial cells at the site of injury	[161]
PHB	–	SD rats with spinal cord hemisection	Promoting differentiation of astrocytes but inhibits their activation	[162]
PHB	–	SD rats with spinal cord contusion	Inhibiting inflammatory bodies, reducing glial scar formation and promoting axonal growth	[163]
PHB	–	SD Rats with spinal cord hemi-contusion	Reducing neuroinflammatory reaction and improving exercise ability	[164]
PSA	MP	SD Rats with spinal cord transection	Inhibiting acute tissue inflammation and apoptosis, and promoting axon regeneration	[165]
PSA	HNK-1	C57BL/6J mice with spinal cord compression injury	Promoting the formation of axonal myelin sheath and improving the recovery of function	[166]
PSA	5-NOT	C57BL/6J mice with spinal cord compression injury	Promoting survival and neurite length of cortical neurons, and improving motor function	[167]
pHEMA	Various substrates and nutritional	Sprague–Dawley rats with spinal cord transection	Increasing the regeneration of the damaged spinal cord and improving motor ability	[168]
pHEMA	–	Wister rats with spinal cord hemisection or transection	Promoting regeneration of axons as well as blood vessels	[169]
pHEMA	–	Sprague–Dawley rats with spinal cord transection	Promoting axonal growth and reducing scar formation	[170]

–: There is no implanted substance

## Data Availability

Not applicable.

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
