# Peer review of "The Application of Biomaterials in Spinal Cord Injury"

_ijms, 2023, doi:10.3390/ijms24010816_

Round 1

Reviewer 1 Report

This is not a good quality review. Too many misses and not up to the mark. Hence, my recommendation is to reject the review paper. 

I detected more than 50% of the texts copied from parts of the published articles while using ithenticate software for plagiarism.

I think the language of the paper is good, it reads well, but I think it could do a lot better content-wise.

For a review, I think the amount of information is inadequate. The authors mentioned most of the common biomaterials in use in the field, but left out many. While it is understandable it is never possible to include everything, I think in this review some major materials are missing as well. In the 'Natural Materials' subsections, I think gelatin, fibrin, acellular matrix derived materials could get their own sections as well given the amount of work that has been done with them.

Although it might not be possible to give every single material its own section, I think the tables could be be made exhaustive at least. For example, there is no mention of many synthetic biomaterials which have been used in the field (hydroxybutyrate, PSA etc.)

The authors have rarely mentioned the drawbacks of the materials they covered. I think the subsections should focus on both the positives and the issues associated with each respective material. (Mechanical strength, bio-resorbability etc.)

I am not sure PLGA is Polylactic acid-ethanolic acid. I am not sure ethanolic acid is even a word. (I might be mistaken on this one).

I think 3D printed materials could have been covered in more detail. A lot of work has been done in the field.

There is no mention of commercially available products, which materials they are derived from and their drawbacks.

The section on nanomaterials seem inadequate to me. Instead of generalizing nanomaterials, I think the authors should have mentioned different metallic nanomaterials, polymer nanomaterials, how they are used, their advantages and drawbacks. The only nanomaterials actually named are graphene and GO. Surely there are many more which have been used with success.

An important paper 'Advances in Biomaterial-Based Spinal Cord Injury Repair' Advanced Functional Materials (2022) is not cited.

Another important review 'Current Concepts of Stem Cell Therapy for Chronic Spinal Cord Injury' by Hidenori Suzuki in this same journal (Intl. J. Mol. Sci) in 2021 has not been cited either! 

Author Response

Response to Reviewer 1 Comments

Point 1: For a review, I think the amount of information is inadequate. The authors mentioned most othe common biomaterials in use in the field. but left out manv. While it is understandable it isnever possible to include everythinq, think in this review some major materials are missincas wel. In the 'Natural Materials' subsections, I think gelatin, fibrin, acellular matrix derivedmaterials could get their own sections as well given the amount of work that has been donewith them

Response: Thank you very much for your scientific comment. We have added seraval descriptions about the Gelatin, Fibrin, and Decellularized scaffold in the “Natural Materials” sections as follows:  

In addition, an increasing number of studies have shown that gelatin, a product of partial hydrolysis of collagen, also has therapeutic effects on SCI. It is a large molecule hydrocolloid that is usually combined with stem cells to form a scaffold to treat SCI. Methacrylate-based gelatin (GelMA) hydrogels are very similar to natural extracellular matrixes (ECM) in some basic properties. One study showed that combining GelMA hydrogels with induced pluripotent stem cell (iPSC)-derived NSCs (iNSCs) significantly promotes the proliferation and facilitates functional recovery of neural stem cells. In addition, GelMA can be fabricated into 3D materials, and loading BMSCs and NSCs into 3D GelMA hydrogels provides excellent mechanical properties for stem cell proliferation, migration, and differentiation. Furthermore, GelMA and acrylated β-cyclodextrin can be formed into a supramolecular bioink (SM bioink), which then loads with an O-GlcNAc transferase (OGT) inhibitor and NSCs. The results showed that this scaffold promotes neuronal regeneration and axonal growth in an injury model. Moreover, a sodium alginate/gelatin 3D scaffold encapsulated with neural stem cells and oligodendrocytes significantly improves hindlimb movement and nerve regeneration. In summary, gelatin and collagen is a promising strategy for SCI repair mainly by combining it with stem cells and many neurotrophic factors while designing GelMA as a 3D or hydrogel scaffold material in terms of its role in the promotion of neural stem cell proliferation, migration, and differentiation. (Lines 290-310, Pages 11)

Fibrin is a protein multimer that is mainly derived from plasma proteins and has good biocompatibility. Injection of fibrin into the SCI can fill the lesion and provide a vehicle for implantation of stem cells, drugs and cytokines. Fibronectin can serve as a carrier for individual peripheral blood nuclei, and its implantation into a porcine SCI model promotes tissue repair in the immediate area of injury and restoration of conduction function in the posterior column of the spinal cord. The fibrin hydrogel prepared by thrombin could smoothly implant nasopharyngeal carcinoma cells into the SCI site by providing cell attachment sites and survival signals and effectively alleviate the immune response at the SCI site. To mimic the natural environment of spinal cord tissue, Yao et al. prepared a 3D hierarchically arranged fibrin hydrogel (AFG) that promotes targeted host cell invasion, vascular system reconstruction and axonal regeneration in SCI rat lesions. In addition, promoting the differentiation of human endometrial stem cells to oligodendrocyte progenitors by upregulating miR-219 expression levels and using fibrin hydrogel as a scaffold to deliver human endometrial stem cells to the site of SCI significantly promoted the recovery of motor function in SCI rats. The use of adipose mesenchymal stem cells combined with fibrin matrix reduces the cavity area, enhances tissue retention, inhibits the activation of astrocytes, and improves the microenvironment at SCI site. He et al. used fibrin hydrogels to deliver exosomes overexpressing VGF, the pro-SCI recovery regulator, and they found that the growth of oligodendrocytes in vivo and in vitro is improved. Therefore, fibrin is usually designed as a hydrogel scaffold, which promotes the regeneration of neural tissue and the recovery of motor function by combining different factors. (Lines 313-337, Pages 11-12)

The decellularized matrix can be made into a porous natural biomaterial scaffold, obtained by removing nerve cells from spinal cord tissue and then freeze-drying them. Decellularized tissue retains the natural ECM components of the spinal cord after decellularization, which contains a large number of signaling molecules that regulate cell development, differentiation, and regeneration. Decellularized tissue removes proteins associated with pathological factors such as the inflammatory response and scar formation and acts as an external microenvironment for implantation into the lesion, optimizing the imbalanced microenvironment after SCI, stimulating axonal regeneration, and promoting injury repair. Cerqueira et al. injected cell-free injectable peripheral nerve matrix (iPN) into the lesions of spinal cord contusions, which successfully established immune tolerance and promoted the survival and axonal growth of SC in the transplanted bodies. Cornelison et al. found that decellularized nerve grafts retain extracellular matrix components such as collagen and glycosaminoglycans, which reduce the ratio of M1:M2 macrophages in a rat cervical contusion model and facilitate axonal growth and nerve tissue repair. The structure of rat spinal cord extracellular matrix scaffold was demonstrated in vitro experiments that the decellular scaffold retains type IV collagen, fibronectin, laminin and other components in a three-dimensional meshwork structure with good pore space, which can promote the attachment and proliferation of bone marrow mesenchymal stem cells. Liu et al. showed that decellularized spinal cord scaffolds can be used for polylactic acid-hydroxyacetic acid copolymer microspheres loaded with the pro-axonal growth drug bisperoxovanadium(pic) (bpV(pic)), which promote the viability of cultured neural stem cells and axonal growth by inhibiting phosphatase gene expression and activating the mTORC1/AKT pathway in vitro, as well as accelerate axonal regeneration and functional recovery in rats with SCI. In addition, decellularized scaffolds can also be used to deliver adipose-derived stem cells (ADSCs), which effectively promotes histopathological repair and axonal regeneration, reduces reactive glial cell proliferation, and promotes functional recovery in SCI rats. Therefore, decellularized scaffolds are often used to improve the internal microenvironment of SCI and reduce macrophage infiltration and cavity area. (Lines 338-369, Pages 12)

Point 2: Although it might not be possible to give every single material its own section, I think the tables could be made exhaustive at least. For example, there is no mention of many synthetic biomaterials which have been used in the field (hydroxybutyrate, PSA etc.)

Response: Thank you very much for your scientific comment. We have added seraval descriptions about the PHB, PSA in the “Synthetic Materials” sections as follows:

Poly-beta-hydroxybutyrate (PHB) is a biopolyester with a large molecular weight that exists in the cytoplasm of many bacteria and is a degradable biomaterial. The transplantation of SCs into PHB catheters could better fill the cavity and significantly increase SC cell attachment, proliferation, and axon regeneration. In recent years, Zhao et al. found that the blending of polyhydroxybutyrate-cohydroxyvalerate (PHBV) with PLA and collagen (Col) could improve the performance of the polymer. The prepared nanofiber scaffold significantly promotes the differentiation of astrocytes but inhibits their overactivation. It can also reduce CSPG and glial fibrillary acidic protein (GFAP) and promotes the recovery of motor function. Another study showed that the implantation of PHBV/PLA/Col membrane and VSC4.1 motor neurons into rats with SCI can inhibit inflammasomes and reduce glial scar formation by reducing the infiltration of CD86-positive macrophages in the lesion and promoting axon growth. In addition, NLRP3 inflammasome expression can be suppressed using the ketone metabolite β-hydroxybutyrate (βOHB), resulting in less neuroinflammation and better motor performance and electrophysiological recovery. Doncel-Pérez et al. used poly(3-hydroxybutyrate-co-3-hydroxycaproic acid) fibrous scaffolds coated with laminin or polylysine/laminin to culture rat neural progenitor cells. Then, synchrotron radiation infrared microspectroscopy observation results showed that the neural progenitor cells had good adhesion and proliferation behavior. Agrawal et al. developed a new type of biomaterial mixed melanin with PHB. This scaffold is compatible with nerve tissue in terms of physicochemical and electrical properties,and has rich surface nanotopography, semiconducting properties, and pore viscoelastic properties similar to brain tissue. Therefore, it can improve cell adhesion and growth of sensory and motor neurons in mice. Therefore, the research on PHB can start from its mechanism of action and suitable composite materials. (Lines 704-731, Pages 21)

Polysialic acid (PSA), a class of linear homogeneous α-2,8 carbohydrates linked to sialic acid, is most prominent during nervous system development and can be combined with neural adhesion molecules in the vertebrate nervous system through typical n-linked glycosidic bonds. PSA can induce the migration of SCs, which has a great effect on improving repair after SCI. Pearse et al. used the lentiviral vector-transduced mouse polysialyltransferase gene ST8SIA4 (LV-PST) and recombinant bacterial enzyme engineering (PSTNm) to upregulate the expression of PSA in nerve cells, proving that PSA can effectively regulate the immune response and improve the migration of SCs and neural repair after SCI. A PSA-based minocycline nanodrug delivery system (PSM) exhibits significant anti-inflammatory and neuroprotective effects in vitro and in vivo, and PSM can recruit endogenous neural stem cells to the injury site in SCI rats, promoting neuronal regeneration and long axon extension throughout the glial scar to improve motor function. Studies have shown that PCL/PSA hybrid nanofiber scaffolds encapsulated with MP implanted into the lesion area can effectively inhibit acute tissue inflammation and apoptosis and promote axon regeneration, thereby promoting nerve repair and improving functional prognosis. In addition, the use of PSA ring mimetic peptide (PR-21) inhibits the hyperplasia of reactive glia and improves motor sensory function. In Mehanna's study, the combination of PSA and human natural killer cell-1 (HNK-1) glycan shows the poor effect in the early stage of injury but can promote axonal myelination and the recovery of function after 6 weeks. Recently, a collagen laminin (C/L) scaffold was used to load the PSA analog compound 5-nonoxytryptamine (5-NOT), which promotes the survival of cortical neurons and neurite length and improves motor function through the ERK-MAPK pathway. In conclusion, finding a suitable PSA mimic is an important direction for future research on SCI. (Lines 732-760, Pages 21-22)

Point 3: The authors have rarely mentioned the drawbacks of the materials they covered. I think the subsections should focus on both the positives and the issues associated with each respective material. (Mechanical strength, bio-resorbability etc.)

Response: Thank you very much for your scientific comment. We have added seraval descriptions in the “Summary and Future Outlook” sections as follows:

At present, SCI has become one of the most difficult central neurological disorders to treat, and patients usually show a decrease or loss of motor and sensory function below the injury site. However, the repair of this damage may be affected by many factors, such as apoptosis of neuronal cells, deposition of glial scarring, and inflammatory cascade response. This article mainly discusses various biomaterial scaffolds for SCI treatment, among which the relatively popular scaffolds are hydrogel and 3D scaffolds. Various materials are designed so that these scaffolds can simulate the internal environment of spinal cord tissue and can meet the repair of the injured part by adjusting the size and shape of the scaffold. Compared with traditional drug intervention and surgical treatment, the use of biomaterial scaffolds can reduce some complex side effects of drugs and obstacles to functional recovery after surgery. The physical and chemical properties of these biomaterials and the structure of the scaffold play an important role in the treatment of SCI. Compared with other treatments, we should pay more attention to the issues of biomaterials, such as biocompatibility, degradability, mechanical strength, and toxicity to peripheral tissues.

Although natural biomaterials mimic ECM and show good biocompatibility, due to their inherent characteristics, there are also some unavoidable shortcomings, such as insufficient scaffold strength, mismatch between cell degradation rate and regeneration rate, and scaffold collapse after the material swells with water. Among them, fibrin has poor mechanical strength and is easy to degrade, which is not conducive to long-term culture in vitro, and the arrangement of fibers lacks directionality, which reduces the induction effect on directional growth of axons. HA is easily soluble in water and absorbs and degrades too quickly. Alginate is prone to the immune response. Synthetic materials increase the hardness of the scaffold but lack inherent biological functions and must undergo significant postprocessing to trigger the desired response in vivo, and their degradation products are prone to local inflammatory responses that destroy the microenvironment, resulting in the decrease in cell survival. In addition, synthetic materials have weak affinity to cells, and which are commonly used to prepare composite scaffolds with natural materials to induce nerve axon regeneration. For example, PCL has insufficient mechanical strength; PLA has a slow degradation rate and poor hydrophobicity; PSA has the disadvantage of being difficult to purify; PHB has poor toughness and a narrow processing time window. It is worth noting that regardless of what kind of biomaterial is implanted into the spinal cord, it cannot cause secondary damage to the body. (Lines 870-902, Pages 24)

Point 4: I am not sure PLGA is Polylactic acid-ethanolic acid. I am not sure ethanolic acid is even word. (l might be mistaken on this one).

Response: Thank you very much for your scientific comment. We modify PLGA to Poly(lactic-co-glycolic acid) (Lines 631-632, Pages 19)

Point 5: I think 3D printed materials could have been covered in more detail. A lot of work has been done in the field.

Response: Thank you very much for your scientific comment. We have added seraval descriptions about 3D scaffold as follows:

3D bioprinting technology allows the size and shape of the scaffold to be adjusted as needed. Many biomaterials are used to design materials as 3D scaffolds for repairing SCI. In a study, a novel biocompatible bioink consisting of functional chitosan, hyaluronic acid derivatives and matrix gels can rapidly gel and spontaneously covalently cross-link to maintain the viability of neural progenitor cells and effectively support axonal regeneration in a rat model of SCI. 3D-printed 3D scaffolds can transport homogeneous bone marrow MSCs and SCs in a specific spatial arrangement, which promote the formation of intercellular junctions and directed cell differentiation. Cryoextrusion 3D printing technology can maintain the biological activity of the delivered cytokines. The collagen/chitosan scaffold prepared by low-temperature extrusion 3D printing technology successfully fuses brain-derived neurotrophic factors, which fills the injury gap, promotes nerve fiber regeneration, accelerates the establishment of synaptic connections, and promotes myelin regeneration at the injury site in rats with SCI. 3D-printed collagen/serin protein scaffolds which mimic the structure of corticospinal tracts can guide the targeted repair of damaged neural tissue in rats. Joung et al. prepared physiologically active neuronal networks by precisely placing clusters of spinal cord neuronal progenitor cells and oligodendrocyte progenitor cells within 3D printed biocompatible scaffolds during assembly and controlling the cell cluster positions using a distribution-by-dot printing method. Koffler et al. used microscale continuous projection printing to create complex neural network structures and printed 3D bionic hydrogel scaffolds loaded with neural progenitor cells that supported axonal regeneration in SCI lesions in mice, induced neuronal synapses upstream and downstream of the implant to connect to neural progenitor cells within the scaffold, and significantly improved functional outcomes. Moreover, the scaffold is scalable to human spinal cord size and lesion geometry. In summary, the 3D scaffold can fill the damaged part well by adjusting its shape and size. (Lines 810-836, Pages 23)

Point 6: There is no mention of commercially available products, which materials they are derived

from and their drawbacks.

Response: Thank you very much for your scientific comment. We have added seraval descriptions about Clinical application of biomaterial scaffolds as follows:

Although many biomaterial scaffolds have been widely used in the study of SCI, few have been translated from preclinical experiments to clinical studies. Previously, Amr et al. investigated the repair effect of chitosan-laminin scaffold-wrapped MSCs on patients with chronic paraplegia. The results showed that motor function is recoveried despite there is a gap in the spinal cord, and the patients regain the ability of some muscle groups, and the thoracolumbar spine is improved. Moreover, the recovery process is not affected by the postoperative seroma formed by the disintegration of chitosan. Xiao et al. studied the safety and feasibility of the collagen scaffold NeuroRegen after implantation into the human body. During the follow-up one year later, they found that there was no adverse reaction related to stent implantation and no more serious consequences such as deterioration of the nervous system. In addition, they observed partial recovery of the patient's autonomic nervous function. In a recent study, after the implantation of MSCs into NeuroRegen scaffolds, the subsequent follow-up found that the sensory function of the patient's bowel and bladder recovered and the sensory level expanded below the injury site. In addition, some patients could control the activities of the toes, suggesting the recovery of some muscle tissue. Chen et al. implanted NeuroRegen stents with autologous bone marrow mononuclear cells (BMMCs) into the body. In addition to stress pulmonary ulcer, pulmonary infection and transient high fever in the short term after the operation, no significant cystic cavitation or malignant proliferation was observed in the follow-up 3 years later. In addition, the patients exhibited expansion of the sensory level and recovery of autonomic nervous function. In addition to the collagen scaffold NeuroRegen, a spinal cord scaffold composed of covalently conjugated poly(lactic acid-glycolic acid) and poly(l-lysine) InVivo Therapeutics Corp and CS collagen scaffolds have also been clinically studied. These clinical studies have confirmed the clinical feasibility of biomaterials, but most of the current clinical trials remain in phase I studies, and the effectiveness of biomaterials in the treatment of SCI still faces great challenges. It is believed that more scaffold materials will be converted into clinical applications in the future. (Lines 837-868, Pages 23-24)

Point 7: The section on nanomaterials seem inadequate to me. Instead of generalizing nanomaterials, I think the authors should have mentioned different metallic nanomaterials, polymer nanomaterials, how they are used, their advantages and drawbacks. The only nanomaterials actually named are graphene and GO. Surely there are many more which have been used with success.

Response: Thank you very much for your scientific comment. We have added seraval descriptions about metallic nanomaterials, polymer nanomaterials in the “Nanomaterials” sections as follows:

In addition to classical graphene nanoparticles, nanoparticles such as metal nanoparticles and polymer nanoparticles also exist. Metallic nanoparticles are currently showing the potential to design novel delivery systems, which can be divided into pure metal nanoparticles and metal oxide nanoparticles. They act on SCI by changing their shape and size, and are then modified by various types of chemical functional groups. Subsequently, modified metal nanoparticles can bind various drugs, antibodies, nutritional factors, etc. Zonisamide (1,2-benzisoxazole-3-methanesulfonamide) has been reported as an antiepileptic drug, but some studies have found that it can exert a certain therapeutic effect on neurological dysfunction. In Fang's study, the use of zonisamide-loaded metal nanoparticles showed promise in modifying neurons and axons to promote recovery from SCI. In addition, metal nanoparticles can also promote the immunogenicity of protein immunity. In one study, gold nanoparticles were used as adjuvants to enhance the activity of a 15 nm GNP-coupled human NgR-Fc (hNgR-Fc) protein vaccine, thereby promoting damage repair. Another study showed that laser exposure of gold nanorods can promote the growth of reaching elements. Therefore, in Mina's study, the authors combined chondroitinase ABCI (cABCI) with different concentrations of gold nanorods. The results of the study exhibited better stability of the enzyme upon binding, thus reducing chondroitin sulfate proteoglycans (CSPG) and promote neuronal regeneration. Moreover, polymeric nanoparticles have also been shown to be therapeutically effective in spinal cord injuries. Ling et al. investigated whether that combining poly(α-lipoic acid)-methylprednisolone (PαLA-MP) prodrug nanoparticles (NPs) and minocycline (MC) could produce better anti-inflammatory effects. In addition, combining rapamycin with mesoporous polydopamine nanoparticles has good ROS clearing ability and exhibits reduced injury cavity, enhanced motility and promotes nerve regeneration in animal models. (Lines 503-528, Pages 15)

Point 8: An important paper 'Advances in Biomaterial-Based Spinal Cord Injury Repair' Advanced Functional Materials (2022) is not cited.

Response: Thank you very much for your scientific comment. We cite the literature in the “Introduction” sections as follows:

In addition, SCI can cause the proliferation of astrocytes and microglia in the nervous system to form glial scars, change the microenvironment of cell survival, increase proinflammatory cytokines, and imbalance the immune response. (Lines 92-95, Pages 2)

Point 9: Another important review 'Current Concepts of Stem Cell Therapy for Chronic Spinal Cord Injury' by Hidenori Suzuki in this same journal (Intl. J. Mol. Sci) in 2021 has not been cited either!

Response: Thank you very much for your scientific comment. We cite the literature in the “Introduction” sections as follows:

Clinical studies have found that stem cells and drugs can be used to improve the SCI microenvironment to treat chronic SCI, and iPSCs introduced into the chronic SCI site can reduce cavitation and support the survival of transplanted cells. (Lines 152-155, Pages 4)

Reviewer 2 Report

-        Add more current references (2021 – 2022).

-        Add also a short information/comparation about metallic alloys.   The following references are suggested: [1] DOI:10.1088/1757-899X/374/1/012023; [2] DOI:10.3390/ma14247610

-        In the Introduction section, the authors cited the specific results of previous research and cited them adequately. However, they did not mention their shortcomings in previous research. In the Introduction section, the penultimate paragraph should contain common features of previous research. The shortcomings of previous research should also be pointed out, in general.

-        In the Introduction section, the last paragraph should contain the scientific contribution and scientific hypotheses of your research. Complete, further elaborate the scientific contribution and scientific hypotheses of your research. Be explicit. In addition to the goal of the research (which was written), the novelty in the context of the scientific contribution should be pointed out. Scientific contributions should be written based on the shortcomings of previous research in the literature. In this way, the authors will better emphasize novelty and scientific soundness.

-        The study also needs to have a discussion where the authors need to discuss their findings and compare them with the previous studies, either review or empirical ones.

-        The discussion should at least conclude something understandable.

-        In the conclusions, state the scientific contribution, the shortcomings of your methodology and future research.

Author Response

Response to Reviewer 2 Comments

Point 1: Add also a short information/comparation about metallic alloys.

Response: Thank you very much for your scientific comment. We have added literature on alloys:

Today, titanium and titanium-based alloys are also widely used biomaterials because of their good biocompatibility, low density, corrosion resistance, and good mechanical efficiency. At present, it is mainly used to replace orthopedic implant materials. The supplement of Zr, V, and Mo elements to titanium can improve the mechanical properties of titanium, which manufactures bone plates and cardiovascular stents. Research has shown that mesenchymal stem cells differentiate into osteoblasts and promote bone union and remodeling when four new alloys are implanted into rabbit models. And titanium-based biomaterials with tailored porosity can affect cell adhesion, differentiation, and growth. Therefore, alloy materials can be considered for the treatment of SCI. When titanium based alloy is used as a biomaterial for SCI repair, it needs to consider that the alloy may react with the liquid environment in the body to produce toxicity and trigger inflammation. Therefore, how to make composite scaffolds from natural or synthetic materials and titanium based alloys is a promising research direction in the future. (Lines 792-809, Pages 22-23)

Point 2: In the Introduction section, the authors cited the specific results of previous research and citedthem adequately. However, they did not mention their shortcomings in previous research. In theIntroduction section, the penultimate paragraph should contain common features of previousresearch. The shortcomings of previous research should also be pointed out, in general.

Response: Thank you very much for your scientific comment. We have added seraval descriptions about the deficiencies of previous research in the "Introduction” sections as follows:

Surgical treatment also has some defects. Patients have to bear postoperative physiological pain, and generated wounds due to surgery can easily damage the physiological environment of the body and cause infection. Early implementation of decompressive surgery has limited efficacy in complete SCI. (Lines 117-121, Pages 3)

Hormonal drugs are prone to adverse reactions, and excessive doses cause other complications. The ingredients of TCS are complex and the mechanism of action is not clear. (Lines 136-138, Pages 4)

Although stem cell transplantation has a wide range of clinical applications, there are problems with stem cell therapy. Cell transplantation may induce lesions in other organs, block blood vessels, induce tumors, and cause rejection in the body, and the mechanism of action of stem cell transplantation is unclear. (Lines 155-159, Pages 4)

Point 3:In the Introduction section, the last paragraph should contain the scientific contribution andscientific hypotheses of your research. Complete, further elaborate the scientific contribution andscientific hypotheses of your research. Be explicit. In addition to the goal of the research (whichwas written), the novelty in the context of the scientific contribution should be pointed out.Scientific contributions should be written based on the shortcomings of previous research in theliterature. In this way, the authors will better emphasize novelty and scientific soundness.

Response: Thank you very much for your scientific comment. We have added seraval descriptions about scientific contribution in the “Introduction” sections as follows:

Compared with other treatment methods, biomaterials are of great value in treating and repairing damaged tissues and assisting in the delivery and release of drugs. Drug delivery through biomaterials can reduce the problems of large invasion and blockage of direct drug administration. Although there are many biomaterials used in SCI in preclinical experiments, there are few generalizations about the characteristics of biomaterials and t review studies about the application of biomaterials in SCI. Therefore, this article reviews various types of biomaterials, focusing on various material properties, treatment modalities, etc. It provides a reference for biomaterials in the clinical treatment of SCI and provide directions for further scientific research. (Lines 165-174, Pages 4)

Point 4:The study also needs to have a discussion where the authorsneed to discuss their findings andcompare them with the previous studies, either review or empirical ones.The discussion should at least conclude something understandable.In the conclusions, state the scientific contribution, the shortcomings ofyour methodology andfuture research.

Response: Thank you very much for your scientific comment. We have added some discussions and conclusions sections as follows:

In conclusion, although these scaffold materials have made some progress in the treatment of SCI, there are still many problems to be solved. When using different scaffold to treat SCI, the detailed mechanism of action still needs to be further studied, and how to combine the drugs, NTs, and stem cells loaded on the stents to achieve the best therapeutic effect still needs to be deeply understood. In addition, when using scaffold for treatment, there is uncertainty about the optimal timing of implantation, location, and duration of treatment while the stent is in the body. In future research, the main research directions include the construction of a stable microenvironment, the study of therapeutic mechanisms, how to improve the shortcomings of various biomaterials, and the development and application of new composite biomaterials in order to find materials with the best therapeutic effect. Most importantly, future research should pay more attention to the safety and effectiveness evaluation of biomaterial scaffolds in clinical trials, so as to promote the application of SCI in clinical treatment. (Lines 903-915, Pages 25)

Round 2

Reviewer 1 Report

Most suggestions and corrections have been taken care of. 

Reviewer 2 Report

Paper was improved.